# S-NPP VIIRS Lunar Calibrations over 10 Years in Reflective Solar Bands (RSB)

**Taeyoung Choi** [1,*]**, Changyong Cao** [2]**, Xi Shao** [3] **and Wenhui Wang** [3]

1   Global Science & Technology (GST), College Park, MD 20740, USA
2   NOAA NESDIS/STAR/SMCD, College Park, MD 20740, USA; changyong.cao@noaa.gov
3   Cooperative Institute for Satellite Earth System Studies (CISESS), University of Maryland, College Park, MD 20740, USA; xi.shao@noaa.gov (X.S.); wenhui.wang@noaa.gov (W.W.)
*   Correspondence: taeyoung.choi@noaa.gov; Tel.: +1-301-683-3562

**Abstract:** Since 28 October 2011, the VIIRS Infrared Imaging Radiometer Suite (VIIRS) on the Suomi National Polar-orbiting Partnership (S-NPP) has operated over 10 years and successfully generated scientific global images for the Earth's environment and climate studies. Besides thermal and day night bands, VIIRS has 14 reflective solar bands (RSBs) that cover a spectral range of 0.41 μm to 2.25 μm. The primary and daily source of calibration for the RSBs is the Solar Diffuser (*SD*) as an onboard calibrator, and its degradations are tracked by the Solar Diffuser Stability Monitor (SDSM). Alternatively, monthly scheduled lunar calibration has provided long-term on-orbit trends that validate the corresponding *SD*-based calibration results. In this paper, on-orbit lunar calibration and comparison results are focused on, in conjunction with the *SD* calibrations that are performed by the National Oceanic and Atmospheric Administration (NOAA) VIIRS team. In addition, a recent study showed that there is increasing striping in the VIIRS images in the RSBs caused by the non-uniform *SD* degradation. The estimation of the *SD* non-uniformity and a mitigation method is proposed along with the striping reductions.

**Keywords:** S-NPP; VIIRS; lunar; solar diffuser; calibration; non-uniformity; scan striping

## 1. Introduction

As of 28 October 2021, the Visible Infrared Radiometer Suite (VIIRS) completed 10 years of successful operation on the Suomi National Polar-orbiting Partnership (S-NPP) satellite [1,2]. The successful launch of S-NPP VIIRS started a new era of earth observations for ocean, atmosphere, land, weather, and other environmental applications. As a successor to historical sensors, such as NOAA's Advanced Very High Resolution Radiometer (AVHRR), National Aeronautics and Space Administration (NASA)'s Terra/Aqua Moderate Resolution Imaging Spectroradiometer (MODIS) sensors, NASA's SeaWiFS instrument, and the Defense Meteorological Satellite Program (DMSP) Operational Linescan System (OLS), VIIRS has continuously provided radiometrically and geometrically calibrated daily global observation data sets called Sensor Data Records (SDR), with 28 derived Environmental Data Records (EDR) [1,3–5] (which are compatible with NASA's Level 1B data). The nominal altitude of the S-NPP satellite is 829 km maintained in an ascending sun-synchronous orbit, with an equator crossing time of 13:30 and an imaging swath width of approximately 3060 km, providing full global daily coverage in both the day and night sides of the Earth [1,2].

VIIRS has 14 Reflective Solar Bands (RSB) that cover a spectral range of 0.41 to 2.25 μm with 11 moderate resolution bands (M-bands), from M1 to M11, and three imaging bands (I-bands), from I1 to I3. The I-band has a nominal spatial resolution of 375 m (371 m by 388 m in along-track by along-scan directions), and the M-band provides a 750 m (742 m by 776 m) resolution [6,7]. In each M-band, there are 16 detectors, whereas each I-band has 32 detectors. To cover a wide dynamic range for different applications, VIIRS adopted

six dual-gain bands that avoid saturation when viewing highly reflective surfaces such as land and cloud targets. In addition, VIIRS uses a pixel aggregation approach to control the gradual pixel growth toward the end of the scan for comparable pixel size within a scan [1,5]. The specifications for the 14 RSB are listed in Table 1, including center wavelengths (CW) and typical radiances (Ltyp and Ttyp) [2,4]. The Relative Spectral Response (RSR) functions are shown in Figure 1, which demonstrates that there are spectrally similar band pairs of I2/M7 and I3/M10.

**Table 1.** VIIRS Center Wavelength (CW), spatial resolution, gain states, and typical radiance for RSB (https://ncc.nesdis.noaa.gov/VIIRS/StandardizedCalibrationParameters.php accessed on 7 July 2022) [2,4].

| Band Name | CW [μm] | Nominal Resolution | Gain States | Ltyp [W/m² sr μm] |
|---|---|---|---|---|
| M1 | 0.410 | 750 m | High/Low | 115/44.9 |
| M2 | 0.443 | 750 m | High/Low | 146/40 |
| M3 | 0.486 | 750 m | High/Low | 123/32 |
| M4 | 0.550 | 750 m | High/Low | 90/21 |
| I1 | 0.637 | 375 m | Single | 22 |
| M5 | 0.671 | 750 m | High/Low | 68/10 |
| M6 | 0.745 | 750 m | Single | 9.6 |
| I2 [#] | 0.861 | 375 m | Single | 25 |
| M7 [#] | 0.861 | 750 m | High/Low | 33.4/6.4 |
| M8 | 1.238 | 750 m | Single | 5.4 |
| M9 | 1.375 | 750 m | Single | 6 |
| I3 * | 1.601 | 375 m | Single | 7.3 |
| M10 * | 1.601 | 750 m | Single | 7.3 |
| M11 | 2.256 | 750 m | Single | 0.12 |

[#] I2/M7 and * I3/M10 are spectrally similar band pairs.

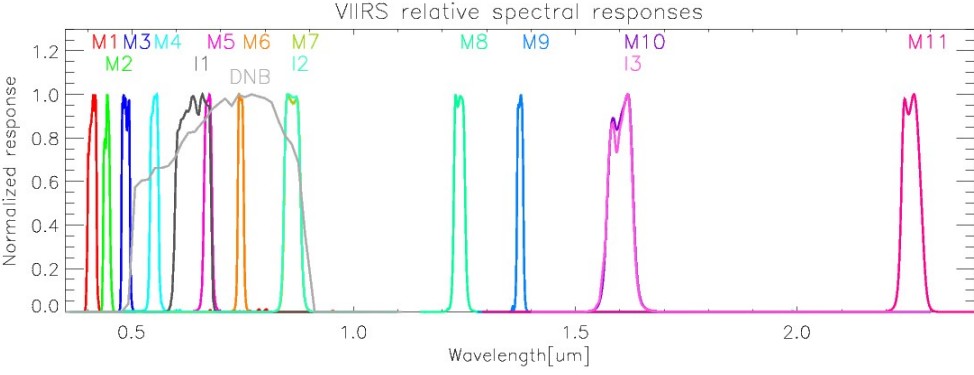

**Figure 1.** S-NPP VIIRS Relative Spectral Response (RSR) functions. The band names are listed above the RSRs in the same color.

The primary source of on-orbit calibration is based on Solar Diffuser (*SD*) observations, and its time-dependent degradation is monitored by the Solar Diffuser Stability Monitor (SDSM). Alternatively, VIIRS can view the moon through the space view (SV) port just before the start of the earth view (EV) port, as shown in Figure 2. The main purpose of the SV port is to provide detectors' zero signal responses, whereas *SD* provides a reflected light source for the RSB radiometric calibration through the *SD* screen [4,8–10]. The *SD* surface properties, called the Bidirectional Reflectance Distribution Function (BRDF), were

carefully measured from the prelaunch calibrations [11], and its functionality was validated by the yaw maneuver data sets as a part of Post Launch Tests (PLTs) in the early life of the sensor [12–14]. Due to the ultraviolet (UV) portion of solar radiation, the surface roughness of the *SD* increases over time once the *SD* is exposed to the sun, and it was modeled as the Surface Roughness Rayleigh Scattering (SRRS) model [15,16]. The time-dependent degradation of the *SD* surface (called H-factor) is monitored by SDSM at eight different wavelengths, with eight detectors inside of a spherical integrating sphere (SIS) [8,9].

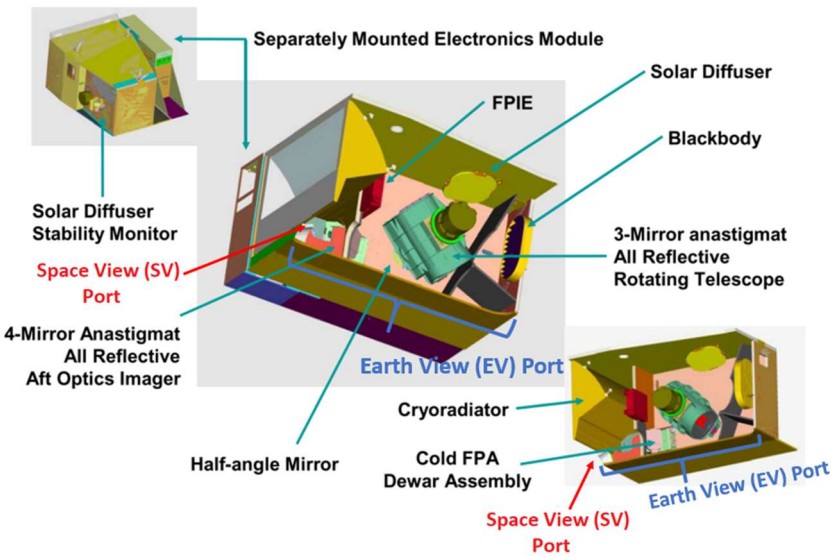

**Figure 2.** VIIRS three-dimensional model of opto-mechanical modules and On-Board Calibrators (OBC) [4].

In conjunction with the H-factor, the *SD* observations have provided on-orbit radiometric calibration coefficients called F-factors from the standardized and official mythology described in the VIIRS ATBD [4]. The VIIRS sensor was designed to have the same Angle of Incidence (AOI) of 60.4 degrees from the Half-Angle Mirror (HAM) at the center of the *SD* and SV ports [17]. By having the same AOI, the calibration coefficients from *SD* and lunar (through SV) observations have the same optical properties at the given AOI. Theoretically, the long-term trends should be matched from the two independent *SD* and lunar F-factors, but they had different long-term trends for *SD* and lunar observations, similar to the lessons learned from the Terra and Aqua MODIS instruments [18–20]. Over 10 years of S-NPP VIIRS operations, there were continuous and consistent growing differences between on-orbit *SD* and lunar F-factors [8,9,21–23]. It was assumed to be caused by the non-uniformity of the *SD* degradation, in respect to the differences between the RTA and SDSM viewing angles [24,25]. However, a recent study showed that the Relative Spectral Response of the SDSM in conjunction with the fast non-linear degradation of the *SD* surface in the short wavelength visible bands (M1–M4) caused the over-estimation of the H-factor, which corrected the long-term degradation differences in the NOAA-20 VIIRS case [17].

The non-uniformity of the *SD* surface was caused by the uneven degradation or illumination of the sunlight for different optical paths. The *SD* non-uniformity over the lifetime of S-NPP VIIRS was identified by the detector level striping in the EV image, with uniform targets such as desert sites and Deep Convective Cloud (DCC) targets [26,27]. The time-dependent *SD* non-uniformity was modeled over the EV targets and applied to the radiometric calibration (in the H-factor) to correct the striping in the EV images that have been applied to NASA's collection 2.0 of S-NPP VIIRS level 1B (L1B) products [26].

In this paper, an overview of the S-NPP VIIRS lunar calibration and its algorithms are provided, along with the *SD* calibration results over 10 years of operation. In the results section, long-term *SD* and lunar calibration trends are compared and validated. In addition,

relative detector response differences between *SD* and lunar observations were measured and correction factors were developed to mitigate striping in the SDR products. Finally, the discussion and conclusion sections suggest challenging issues, future improvements, and a summary of this paper.

## 2. On-Orbit Radiometric Calibration Algorithms

### 2.1. Earth View (EV) Top-of-Atmosphere (TOA) Radiance

As a baseline product, VIIRS SDR provides TOA radiance or reflectance values in each pixel, with quality flags and metadata in the RSB bands in a Version 5 Hierarchical Data Format (HDF5) format [5]. As shown in Equation (1), the *EV* radiance is calculated by the offset-corrected DN that is indicated by using lower-case letters $dn_{EV}$, c-coefficients (as shown $c_0$, $c_1$, and $c_2$), and emphSD F-factor ($F_{SD}$) and by dividing Response Versus Scan (*RVS*) at the corresponding angle of incidence on the Half-Angle Mirror (HAM) at the specific *EV* angle.

$$L_{EV} = \frac{\left(c_0 + c_1 dn_{EV} + c_2 dn_{EV}^2\right) F_{SD}}{RVS_{EV}} \tag{1}$$

The *EV* radiance is calculated in each band, detector, sample, and scan. In the dual gain bands, the different gain state is also considered in the *SD* F-factor and c-coefficient. The mean level of the quadratic terms $c_2/c_1$ term was very small on the order of $10^{-6}$, indicating that there was a very small non-linearity with the RSB detectors [11]. To compensate for time-dependent sensor responsivities, the *SD* F-factors were automatically updated and applied to the NOAA's operational Interface Data Processing Segment (IDPS) production system, which generates VIIRS data products in near real-time with a latency of a few hours [28].

### 2.2. SD Calibration

The primary source of calibration is from the *SD* observations. When the S-NPP satellite goes from the night side to day side near the South Pole, the *SD* panel obtains ample illumination from the sun through the *SD* attenuation screen. Usually, there are 14 or 15 opportunities of proper *SD* illumination per day. At the same time, SDSM is also illuminated through a separate screen, as shown in Figure 3. The geometric orientation of the sun continuously changes toward the *SD* surface, and there are desirable conditions for proper *SD* illumination, which is called the 'sweet spot' [4,9,29]. Within the sweet spot, the *SD* F-factors are calculated from Equation (2):

$$F_{SD}(t) = \frac{cos\theta_{inc} \int RSR \, \Phi_{Sun} d\lambda \, \overline{\tau_{SDS} BRDF_{SD}(t)} RVS_{SD}}{4\pi d_{Sun}^2 \int RSR \, d\lambda (c_0 + c_1 dn_{SD}(t) + c_2 dn(t)_{SD}^2)} \tag{2}$$

where $\theta_{inc}$ is the incoming solar angle to the *SD* surface, $\Phi_{Sun}$ is the solar spectral power as a function of wavelength ($\lambda$), $\overline{\tau_{SDS} BRDF_{SD}(t)}$ is the screen transmittance function and *SD* BRDF function at the time of observation, and $RVS_{SD}$ is *RVS* at the *SD* angle, which is close to unity. The $c_0$, $c_1$, and $c_2$ are called prelaunch c-coefficients that convert bias-removed *SD* $dn$ ($dn_{SD} = DN_{SD} - DN_{SV}$) to radiance from the quadratic equation, and $d_{Sun}$ is the distance between the satellite and the sun.

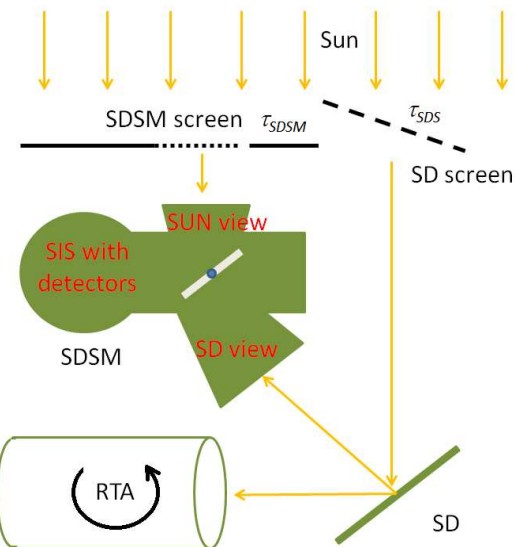

**Figure 3.** A simplified RTA, SDSM, *SD*, and RTA diagram with *SD* and SDSM screens. The SDSM has a rotating mirror to select SUN, *SD*, or dark view responses with the detectors in the Spherical Integrating Sphere (SIS) [30].

*2.3. SD Degradation Estimation from SDSM*

The surface roughness of the *SD* has been growing over time because it has been exposed to the solar ultra-violet radiation when the sensor became space-borne [15,16]. Figure 3 shows a simplified schematic of the relationship among *SD*, SDSM, and RTA, along with the pathways of solar illumination. The *SD* degradation, also called the H-factor, is monitored by SDSM as a ratioing radiometer from the sun and *SD* observations. The H-factor is calculated by Equation (3):

$$H(t) = \frac{dc_{SD}\ \tau_{SDSM}}{dc_{SUN}\ cos\theta_{inc}\ BRDF_{SDSM\_SD}\ \tau_{SDS}\ \Omega_{SDSM}} \tag{3}$$

where $dc_{SD}$ and $dc_{SUN}$ represent the bias-removed Digital Count (DC, named to differentiate from the DN for *SD* case), $\tau_{SDSM}$ is the *SDSM* sun screen transmittance, $cos\theta_{inc}$ corrects the cosine effect of incoming light, $BRDF_{SDSM\_SD}$ is *BRDF* in view of the *SDSM SD* port, and $\Omega_{SDSM}$ is the solid angle correction of the *SD* view port. Please note that DC is the unit for *SDSM* detectors, whereas DN is the unit for RSB detectors. When the H-factor is applied to the *SD* F-factors, it is normalized to the initial point, as shown in Equation (4). Equation (4) is normalized at zero-time stamp $BRDF_{SD}(0)$, and $H\_factor(0)$ represents an initial H-factor, respectively. The time-dependent H-factor is applied to modify the *SD* *BRDF*, and it linearly affects to *SD* F-factor, as shown in Equation (2).

$$BRDF_{SD}(t) = \frac{H\_factor(t)}{H\_factor(0)} BRDF_{SD}(0) \tag{4}$$

*2.4. Lunar Calibration Using Scheduled Lunar Collections*

2.4.1. Lunar Irradiance Model

The radiometric stability of the moon surface has been widely accepted within the visible and shortwave spectrum ranges [31–34]. On-orbit lunar calibration requires an accurate lunar irradiance model to be used as a radiometric calibration reference, which needs to be compared to the observation from an imaging sensor in situations of constantly changing geometric conditions among the sun, moon, and sensor. The Robotic Lunar Observatory (ROLO) model was developed by the US Geological Survey (USGS). The ROLO lunar irradiance model accounts for the dependence of lunar irradiance on geometric variables, such as the sun–Earth and moon–Earth distances, lunar phase angle (angle

between moon–sun and moon–Earth vectors), and the libration angle [9]. Using more than 10 years of lunar observations, the ROLO model was developed from ground telescope-based moon collections at 32 wavelengths between 350 and 2450 nm, within an absolute lunar phase angle limit between 2 and 92 degrees [35]. The overall uncertainty level of the ROLO model should be less than one percent in most of the RSB bands [9]. The European Organization for the Exploitation of Meteorological Satellites (EUMETSAT) implemented the ROLO model as a standard lunar calibration tool for the GSICS community in collaboration with USGS, NASA, Japan Aerospace Exploration Agency (JAXA), and Centre National d'Etudes Spatiales (CNES) [9,29,35].

2.4.2. Procedures of Lunar Collection

As shown in Figure 4, VIIRS can view the moon on a monthly basis through the SV port within a scan range from −66.1 to −65.25 degrees before the start of the EV scan. To locate the moon at the center of the SV port, a spacecraft roll maneuver needs to be predicted and planned ahead of the desired time of the scheduled lunar collection. Approximately one month before the lunar collection, the NOAA VIIRS team collects orbital information (called ephemeris file) and predicts candidate dates, times, spacecraft roll angles, and lunar phase angles using the NASA Jet Propulsion Laboratory's (JPL) Navigation and Ancillary Information Facility (NAIF) tool (https://naif.jpl.nasa.gov/naif/index.html accessed on 7 July 2022).

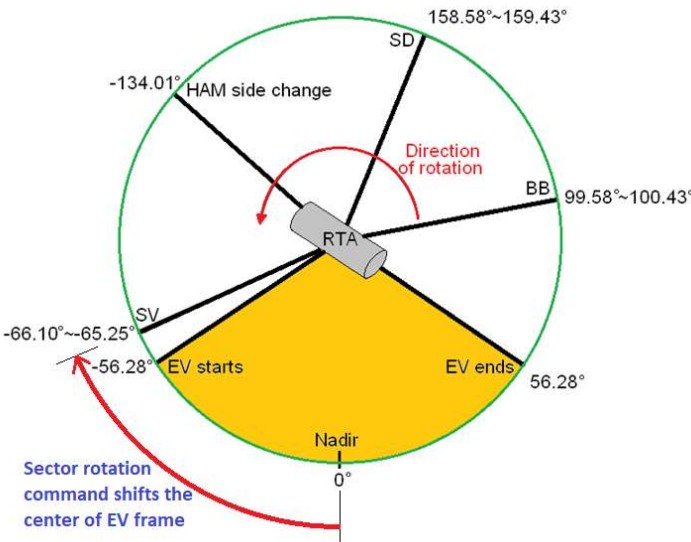

**Figure 4.** S-NPP VIIRS RTA scan angle at EV, BB, *SD*, and SV views [4]. The moon can be viewed through the SV port. The sector rotation command will place the center of the EV frame at the middle of SV range. This will be explained in Section 2.4.3.

There are some satellite operational constraints when the predictions are calculated. The lunar phase should be within −51.5 to −50.0 degrees (negative phase angle means waxing moon), which is a user side requirement for the lunar collection to reduce the uncertainties having variations in the lunar phase angles. Another condition is that the spacecraft roll angle should be between −15 to 1 degree. The South Atlantic Anomaly region is also avoided in terms of orbital location when the lunar roll maneuver is performed.

Once the desired times of the lunar roll maneuver are calculated, the best one is selected and validated with the NASA VIIRS Calibration Support Team (VCST). The selected lunar roll time and roll angle information are notified to the NOAA Satellite Operational Facility (NSOF), and the command is communicated to the satellite and then the lunar collection is executed. Figure 5 shows the detailed procedures of a scheduled lunar collection.

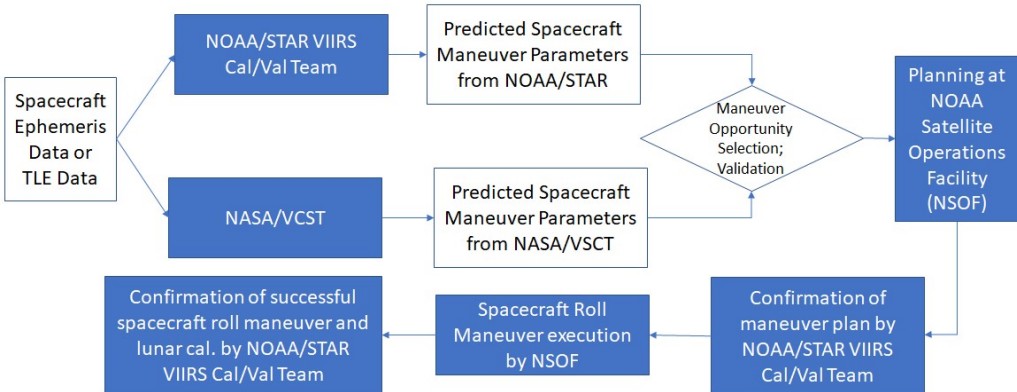

**Figure 5.** VIIRS scheduled lunar collection planning, prediction, decision, execution, and validation procedure diagram.

The lunar roll maneuver is a mission-critical activity that affects all other sensors on the S-NPP satellites, such as Advanced Technology Microwave Sounder (ATMS), Ozone Mapping and Profiler Suite (OMPS), Cross-track Infrared Sounder (CrIS), and Clouds and the Earth's Radiant Energy System (CERES). Lunar prediction is challenging because the roll maneuver timing window and spacecraft roll angle need to be accurately determined down to one second and 0.01-degree level when it is notified to NSOF. After the desired date, the NOAA VIIRS team provides confirmation of the successful spacecraft roll maneuver and lunar collection by checking the SDR granules near the predicted date and time, as shown in Figure 6. It should be noted that the VIIRS is fixed to be in high gain mode for the dual-gain bands because of the low radiance level of the moon.

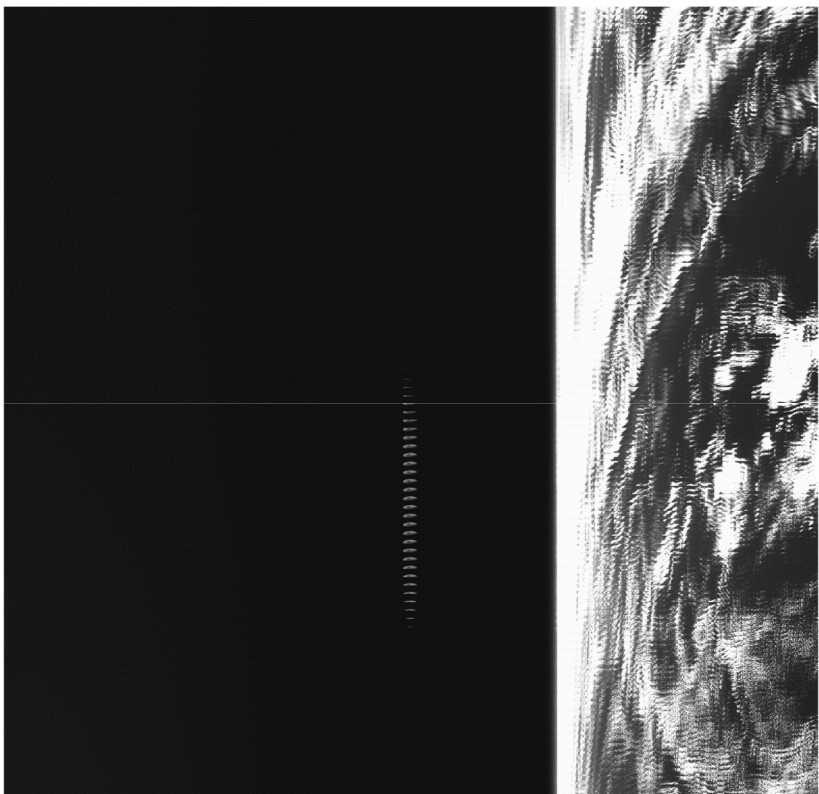

**Figure 6.** S-NPP VIIRS lunar calibration data were acquired on 14 November 2021 18:43 and 18:45 UTC Raw Data Record (RDR) granules for band M7. The white line in the middle indicates a granule dividing point between 18:43 and 18:45 UTC. The moon is placed at the center of the EV frame with the proper lunar roll maneuver and sector rotation command.

### 2.4.3. Lunar Calibration

After performing the lunar roll maneuver, lunar calibration starts with collecting the lunar Raw Data Record (RDR) granules near the center time of lunar collection, as shown in Figure 6. For the S-NPP VIIRS case, the active lunar scans can be identified at the center of the EV frame with the accurate roll maneuver and sector rotation command uploaded to the VIIRS sensor. The sector rotation command shifts the SV frame to be at the center of the EV frame during the lunar roll maneuver [29]. By applying the sector rotation, the RTA angle coverage of 0.85 degrees at the SV port becomes expended to the full extent of the EV angle range of 112.56 degrees, as shown in Figure 4. In each scheduled lunar collection, there are multiple scans of the moon at the center of the EV frame, as shown in Figure 6. From the center of the EV frame, the bias removed DN (or detector zero signal offset) is calculated from either side of the moon in each band, detector, frame, and scan. Before aggregating for irradiance, the lunar pixels are converted to radiance using EV radiance Equation (1). For lunar calibration, the RVS becomes unity at the SV angle and the *SD* F-factor is fixed at the first S-NPP VIIRS lunar collection time at 147 Days Since Launch (DSL). After the conversion to radiance, the observed lunar irradiance and lunar F-factor are calculated from Equation (5) [29].

$$F_{Lunar}(t) = \frac{I_{GIRO}(t)}{I_{OBS}(t)} = \frac{I_{GIRO}(t)}{\sum_{Pixel} \frac{L_{Pixel}(b,d,f,s,t)}{N(t)} \frac{\pi \cdot R_{moon}^2}{D_{Sat\text{-}Moon}^2(t)} \frac{1 + \cos\theta(t)}{2}} \tag{5}$$

The lunar F-factor is a ratio between lunar irradiance from the Global Space-based Inter-Calibration System (GSICS) implementation of the Robotic Lunar Observatory (ROLO) (*GIRO*) model [9] and observed lunar irradiance ($I_{OBS}$), which can be calculated by aggregating all the lunar radiance values in all the effective pixels. The second term's denominator of Equation (5) represents the full solid angle of the moon at the time of lunar collection by the moon radius ($R_{moon}$) and the distance between the satellite and moon ($D_{Sat\text{-}Moon}$). The third term represents the actual effective portion of the moon affected by the phase angle ($\theta$).

### 2.5. SD Non-Uniformity Estimation Using Lunar Observation

One of the effective methods used to evaluate the quality of SDR products is checking the EV imagery with a uniform target such as Deep Convective Cloud (DCC) and Pseudo Invariant Calibration Sites (PICS) [36–38]. Because of the increasing non-uniformity of the *SD* surface among the detectors, the VIIRS SDR images observed in the short wavelength range show the track direction striping especially at the neighboring scan boundaries [26,27]. The striping gradually increased with time when the images were tested in recent years. Figure 7 is the S-NPP VIIRS SDR image over the Red Sea region on 5 May 2021 in band M1. Scan-related striping is evident in the Red Sea area (in the box) and in the uniform area in the land.

S-NPP VIIRS striping in the short wavelength bands (M1 to M4) is caused by the non-uniform degradation of the *SD* surface that affects biases in the *SD* F-factors in the track direction (or detector array direction) of *SD* observations. Figure 8 shows detector arrays in the visible and near-infrared (VIS/NIR) Focal Plane Assembly (FPA) from the VIIRS geolocation Algorithm Theoretical Basis Document (ATBD) [6]. After ten years of on-orbit UV exposure, the *SD* surface had degraded approximately 44 percent of its reflectance at the shortest wavelength of SDSM detector 1 at 0.412 μm. With this large amount of *SD* surface reflectance degradation, there is no guarantee that it had been degraded uniformly in the detector array direction. On the other hand, there are higher chances of having differences of *SD* degradation in different detector positions, especially between the first and last detector, considering the mounting location of the instrument, as shown in Figure 8.

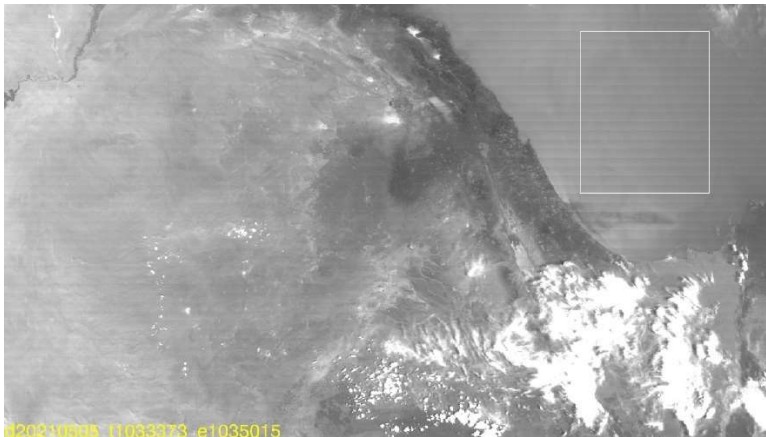

**Figure 7.** S-NPP VIIRS SDR image in band M1 near the Red Sea region on 5 May 2021. The image timestamp is also shown at the lower left corner. Scan-related striping is evident in the Red Sea area in the box and over the land.

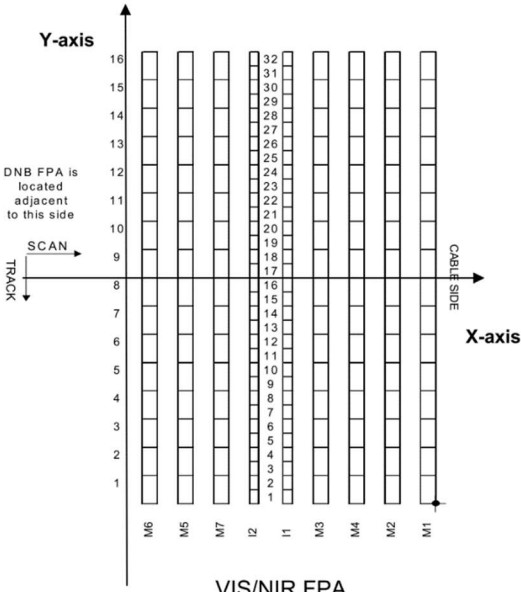

**Figure 8.** S-NPP VIIRS visible and near-infrared (VIS/NIR) Focal Plane Assembly (FPA) detector layout, along with scan and track direction indicators.

The non-uniform *SD* degradation can be estimated by measuring the detector response differences between the *SD* and lunar collection under an assumption that all the VIIRS detectors view the same lunar surface in the multiple scans in Figure 6. According to the detector layout in Figure 8, the lunar radiance image was reorganized to the detector view of the moon for all the scheduled lunar collections. Figure 9 shows the image conversion from the scan-based collection to the detector view in bands M1 to M4.

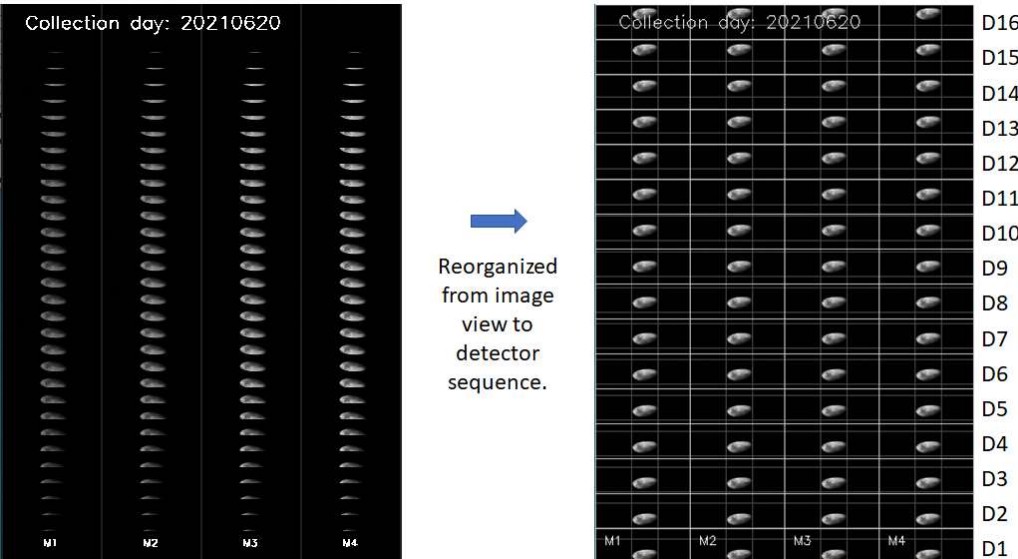

**Figure 9.** Lunar image conversion to detector sequence in each scan, and detectors from the scheduled lunar collection on 20 June 2021 in bands M1 to M4. The white lines are the dividing band (and detectors) and the gray lines on the right image indicate the moon area box to reduce the noise from the outside of the box in the detector images.

The detector response difference was calculated from the detector-dependent lunar irradiance using all the scans in each lunar collection, as shown in Figure 9. The basic assumption was that the overlap (or underlap) among the scans were identical to all the detectors, and the different samplings among the detectors were assumed to be very similar. In each detector image, the relative location of the moon changed from the bottom of the frame box to the top side along with the detector numbers. The time-dependent lunar detector differences were fitted and modeled by following Equation (6). The retrieved TOA detector lunar irradiance can be derived as

$$Det\_Diff_{Lunar}\ (t,d) = \frac{Irrad_{Lunar}(t,d) - \overline{Irrad_{Lunar}(t)_{det}}}{\overline{Irrad_{Lunar}(t)_{det}}} = d_{L,0} + d_{L,1}t \qquad (6)$$

Where $\overline{Irrad_{Lunar}(t)_{det}}$ indicates the averaged detector irradiances across all the detectors in a band, and $d_L$ represents the linear fitting coefficients over time. For the lunar irradiance trends, a linear fit was applied to reduce the noise in the lunar response differences. Details of noise reduction procedures will be discussed in the results section.

At the same time, *SD* detector radiance differences are also derived similarly by Equation (7). Please note that the unit of *SD* BRDF was radiance, according to its definition. The time-dependent *SD* detector radiance difference, $Det\_Diff_{SD}\ (t,d)$, was not modeled but interpolated from the finely sampled measurements because the noise in the *SD* detector level radiances is very low.

$$Det\_Diff_{SD}\ (t,d) = \frac{Rad_{SD}(t,d) - \overline{Rad_{SD}(t)_{det}}}{\overline{Rad_{SD}(t)_{det}}} \qquad (7)$$

Figure 10 shows example results between the normalized lunar and *SD* detector difference profiles in band M1 and detector 16. The red line indicates the lunar irradiance difference in detector 16 that increased approximately up to a one-percent level, whereas the *SD* detector radiance response went up almost two percent. The ratio of these two differences was calculated by Equation (8), characterizing the lunar and SD calibration differences in the detectors. It was applied back to EV SDR products to mitigate the striping

in the scan direction by Equation (9). The $\hat{d}$ indicates product order of the detector, which is opposite of the calibration order for the rest of the detector sequence in other equations.

$$Det\_Cor(t,d) = \frac{Det\_Diff_{Lunar}(t,d)}{Det\_Diff_{SD}(t,d)} \tag{8}$$

$$L_{EV_{Cor}}\left(t,\hat{d}\right) = L_{EV}\left(t,\hat{d}\right)Det\_Cor\left(t,\hat{d}\right) \tag{9}$$

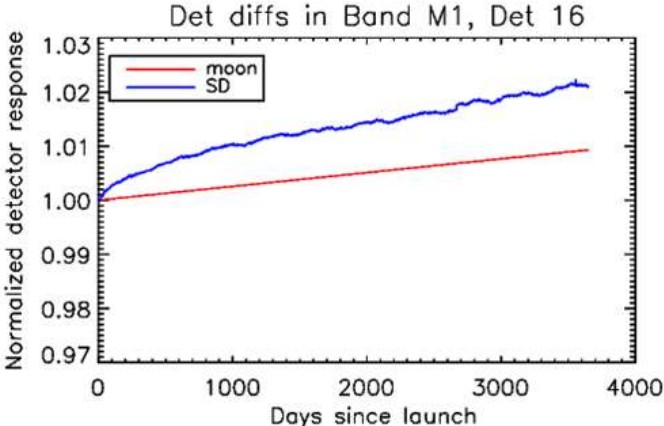

**Figure 10.** Normalized lunar and *SD* detector difference plot. Because of the increasing non-uniformity on the *SD* surface over time, the SD detector difference increased up to a two percent level, whereas the lunar detector difference suggested one percent around Days Since Launch (DSL) 3500. Please note that the detector sequence in this image is in the calibration order, which is the opposite of the product order.

## 3. On-Orbit Calibration Results

### 3.1. SD Degradation (H-Factor)

To measure on-orbit SD degradation, the SDSM was operated in every orbit initially, and it was reduced to weekly, along with its consistent long-term trends. By operating weekly, the expected lifetime of SDSM was assumed to be extended by saving mechanical movements. As of December 2021, there were more than 2400 SDSM collections that had been calculated and applied to S-NPP VIIRS RSB calibration. Figure 11 shows raw H-factors in symbols, and the fitted trends are shown as black solid lines for all the eight SDSM detectors. For SWIR band calibration, the H-factors were assumed to be one for current operational products.

As shown in Figure 11, the raw H-factors showed reasonably stable responses over time, with some degree of oscillations starting from 2014 to the middle of 2015. The source of oscillation is unknown, but during the period, the satellite orbital characteristics were not the same as the rest of its life. From the start of 2014 to the end of 2015, S-NPP's Local Time Ascending Node (LTAN) came up to 13:34 compared to the nominal operational time of 13:25. In addition, the solar azimuth angle range was slightly shifted around two degrees, as shown in Figure 12. These different geometric conditions may have altered solar illumination conditions on the SD surface, and it probably changed the rate of degradation during the time.

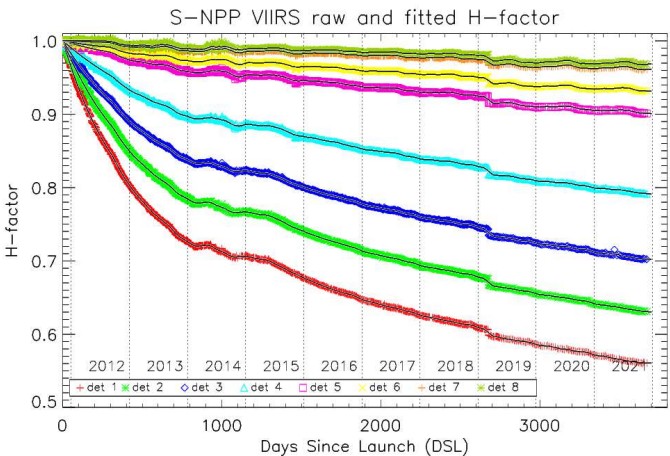

**Figure 11.** S-NPP VIIRS SD degradations (H-factors) over ten years. The vertical dotted lines represent year division lines.

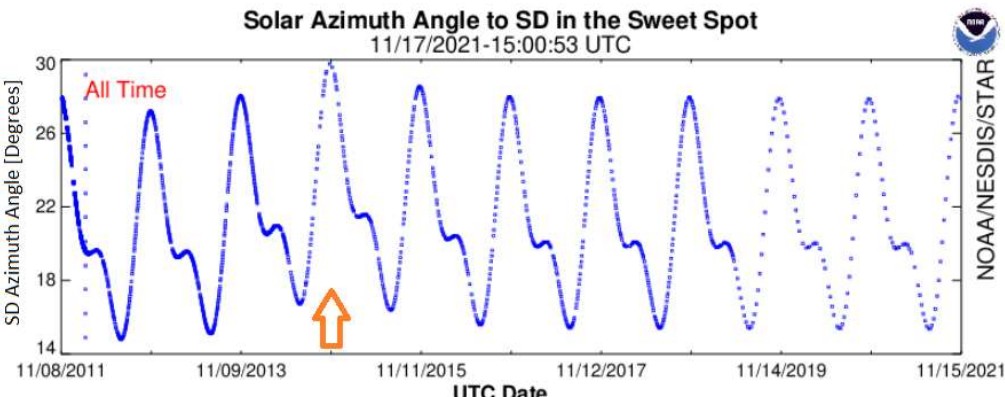

**Figure 12.** S-NPP VIIRS solar azimuth angle to SD in the sweet spot. The solar azimuth angle range was shifted up during the H-factor oscillation period near 2014, as indicated by the orange arrow. This image is taken from NOAA Integrated Calibration Validation System (ICVS) webpage at https://www.star.nesdis.noaa.gov/icvs/status_NPP_VIIRS.php (accessed on 15 November 2021).

On 24 February 2019, there was a sudden one percent level drop in the H-factor as shown in Figure 11. It happened not only for the H-factors, but it also affected the *SD* F-factors. After the anomaly, the NOAA VIIRS team investigated the SDSM data and found that the *SD* signal was suddenly dropped in SD and SDSM observations to the SD surface. The root cause of this anomaly is also unknown, but it is assumed that a small portion of the *SD* screen was blocked either on the *SD* screen or *SD* surface, reducing the radiance from the *SD* surface. The effects of the *SD* F-factors will be discussed in the following subsection.

Besides these two anomalies, S-NPP VIIRS H-factors reasonably represented on-orbit *SD* surface degradations ten years at the assigned wavelengths. As defined in Equation (4), the eight trends are all normalized to the initial point when the instrument is launched. The *SD* surface has gradually degraded over time, especially toward short wavelengths from SDSM detector 1. After ten years of exposure to the sunlight, *SD* reflectance has degraded approximately 44 percent in detector 1.

### 3.2. SD F-Factors

In each orbit, the *SD* has a chance to be fully illuminated when it goes into daytime near the South pole. Within the desirable illumination conditions called the 'sweet spot,' the F-factors are calculated in each band, detector, gain state, and HAM side. Usually, there are 14 or 15 orbits per day, and these individual *SD* F-factors are averaged daily. Since the

*SD* calibration started on 8 November 2011, with orbit number 154, there were 51,391 *SD* F-factors (or orbits) as of 8 November 2021, after a decade. Figure 13 shows the band averaged SD F-factors. The SD F-factors (inverse of gain) near the wavelength of 1.0 μm, such as M7, M8, and I2, showed large F-factor increases (or detector gain degradations), up to 80 percent increases (or 45 percent gain loss, approximately). The main source of large degradation in these bands was due to the increasing Tungsten contamination on the RTA mirror surface, which darkened the surface centering near 1.0 μm, affecting the *SD* F-factors and the Signal to Noise Ratios (SNRs) [39–41]. However, the Relative Spectral Response (RSR) changes to the *SD* F-factors were very minimal in all RSB bands that were less than 0.12 percent levels after 2 years of operation.

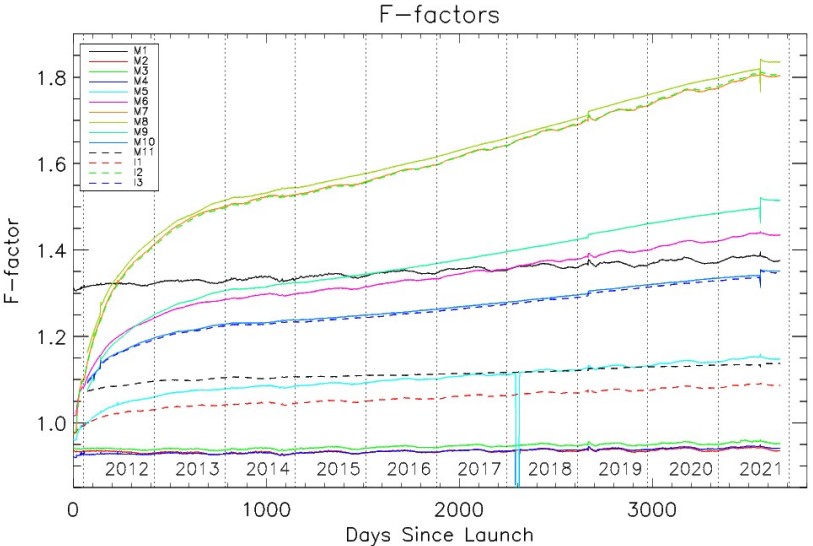

**Figure 13.** S-NPP VIIRS *SD* F-factors over ten years. The vertical dotted lines represent year division lines.

In the short wavelength side, the *SD* F-factors were very stable and showed steady but small increases over time in bands M1 to M4. In all these short-wavelength bands, the SD F-factors showed very stable long-term decadal trends, with 5%, 0.5%, 1.5%, and 2.5% for bands M1, M2, M3, and M4, respectively.

The sudden drop in band M4 and M5 on 15 February 2018 was caused by a mission operation error. During DNB calibration upload to the instrument, there was a data acquisition schedule error and it was corrected on 7 March 2018.

Another sudden spike on 24 February 2019 was already explained in the H-factor result Section 3.1, and it was caused by *SD* and SDSM signal drops. Initially, the *SD* F-factors were slightly fluctuated but returned to near nominal levels, as shown in Figures 13 and 14, when the updated H-factors were applied in bands M1 to M7, I1, and I2. Because of the time-delays in the H-factor filtering (in Figure 11) around 24 February 2019, there were sudden F-factor spikes with the immediate *SD* DN signal changes until the H-factor went back to the nominal trends. However, there were slight sudden increases in the SD F-factors (up to 0.5%) in SWIR bands (I3 and M8~M11) since these bands were not adjusted by the H-factors.

On 3 August 2021 at 12:46 UTC, all the S-NPP instruments, including VIIRS, went into safe mode when re-enabling the star trackers as a part of the star catalog uploads. It came back to the nominal mission pointing status at 22:28 UTC on the same day. NOAA and NASA VIIRS teams analyzed the impacts to the product and found that there were no significant impacts to the RSB bands. The *SD* F-factors had slight spikes in all the RSB bands and came back to nominal trends. However, there were slight *SD* F-factor increases in the SWIR bands, as shown in Figure 13. On the other hand, there were no observable Signal to Noise Ratio changes due to the safe mode anomaly in RSB.

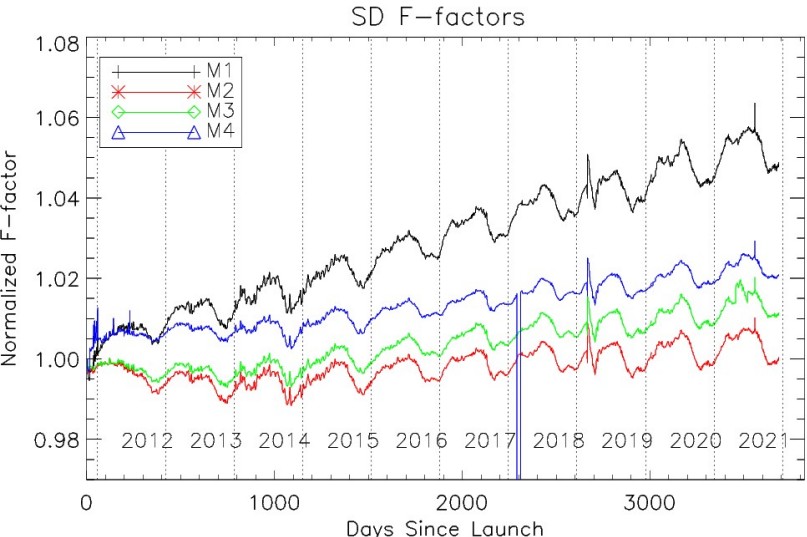

**Figure 14.** S-NPP VIIRS normalized *SD* F-factors from M1 to M4. These short-wavelength bands showed stable responses over ten years.

### 3.3. Lunar and SD F-Factors Comparison Results

The lunar F-factors are calculated by Equation (5), which represents the ratio of GIRO lunar irradiance and the VIIRS observed lunar irradiance. In an ideal condition, the lunar F-factors should be near unity (similar to the SD F-factor case); however, there were static offsets in the long-term trends, mostly within ±8 percent levels, according to our previous study [9]. In Figure 15, the best fitting scaling factors that minimize the difference between the *SD* and lunar F-factors were found and applied to the lunar F-factors (symbols) to be on the *SD* F-factors (lines) in Figure 15. In all the RSB bands, the *SD* and lunar F-factors showed reasonable long-term agreements within 1.8 percent standard deviation level of the ratio differences between *SD* and lunar F-factor points. Table 2 shows the standard deviations of the differences between lunar and *SD* F-factors and the best fitting scaling factors.

The difference STD values increased recently in bands M6, M8, and I1 because of the larger difference in early 2021. With these two-roll maneuver-free collections, the location of the moon was closer to the earth limb than usual. That probably increased the earthshine near the moon, which increased the radiance levels of the lunar observations, especially in some NIR bands. The best-fitting scaling factors in Table 2 are mostly near unity, which compensated absolute scale differences caused by solar irradiance model differences between GIRO and VIIRS *SD* F-factors.

The *SD* and lunar F-factors showed reasonable long-term agreements in RSBs, but there were growing differences, especially in the short wavelength bands, as shown in Figure 16. To effectively visualize these differences, the *SD* F-factors were normalized in a zoomed-in y-scale. During the early years of operation, the lunar F-factors were quite lower than the *SD* lines when they were compared to the late years. To correct these long-term trend differences, the NOAA VIIRS team developed a comprehensive correction method not only using lunar F-factors but also including Deep Convective Cloud (DCC) and Simultaneous Nadir Observation (SNO) [23]. On the other hand, the NASA VIIRS and NOAA ocean color teams developed and applied their best practices by using *SD* and lunar observations only [9]. Compared to the *SD* calibration, there were disadvantages of lunar calibration. The number of monthly lunar collections was too small compared to the orbit-based *SD* F-factor with large annual oscillation levels. To mitigate these problems, other long-term calibration sources such as DCC, SNO, and inter-satellite calibration results were strongly suggested, such as the Kalman filtering method that the NOAA VIIRS team developed and applied [23].

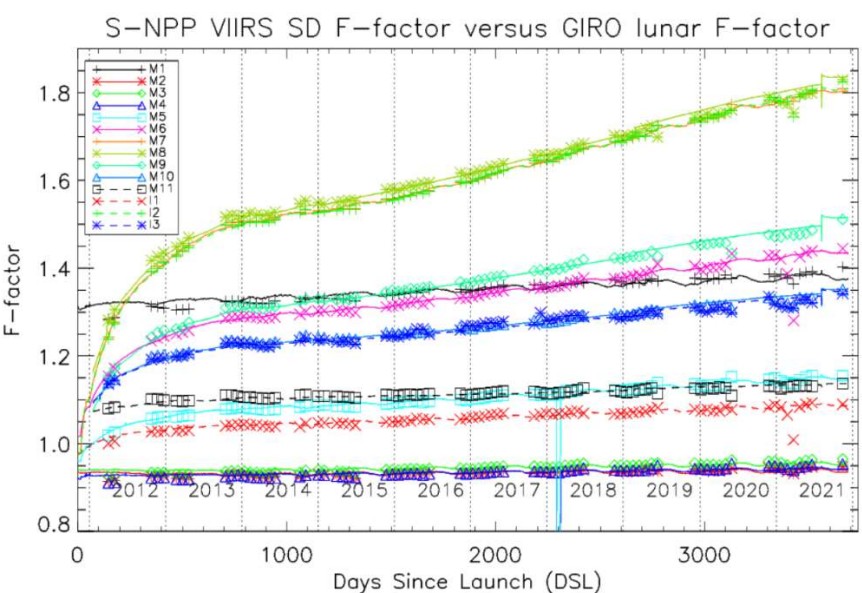

**Figure 15.** S-NPP VIIRS long-term *SD* and lunar F-factor comparison results. The lines are *SD* F-factors and the symbols are lunar F-factors. The lunar factors were normalized to the *SD* F-factor trends with best-fitting factors.

**Table 2.** S-NPP VIIRS lunar F-factor standard deviations (STD) of the difference to the *SD* F-factors and best fitting scaling factors (SF).

| Band | M1 | M2 | M3 | M4 | M5 | M6 | M7 | M8 | M9 | M10 | M11 | I1 | I2 | I3 |
|------|------|------|------|------|------|------|------|------|------|------|------|------|------|------|
| STD% | 0.16 | 0.60 | 0.54 | 0.58 | 0.41 | 1.87 | 0.45 | 1.16 | 0.82 | 0.65 | 0.48 | 1.02 | 0.79 | 0.94 |
| SF | 1.020 | 0.970 | 1.031 | 0.992 | 0.988 | 0.913 | 0.936 | 0.882 | 0.915 | 0.960 | 1.064 | 1.037 | 0.977 | 1.064 |

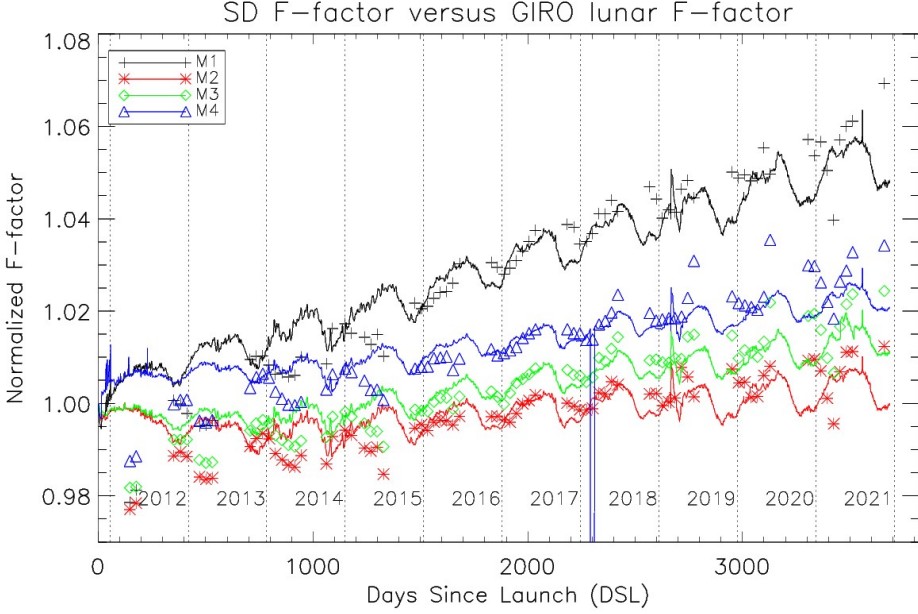

**Figure 16.** Normalized *SD* and lunar F-factor comparison results in bands M1 to M4. The *SD* F-factors were normalized on the first day of *SD* collection on 8 November 2011. The lines are *SD* F-factors and the symbols are lunar F-factors.

### 3.4. Estimated Detector Response Difference using SD and Lunar Collections

3.4.1. Lunar Detector Response Differences

As explained in Section 2.4, the lunar detector response differences were calculated by Equation (6). In each detector, the normalized detector differences were noisy, as shown in Figure 17. There were extreme outliers, and these points (gray lines) were removed by rejecting points larger than ±5 percent levels. After the rejection filtering, the lunar detector differences were fitted with a linear function. The lunar detector differences were further filtered by selecting data sets within ±1 standard deviation from the fitted line, shown as red diamonds and a line in the figure.

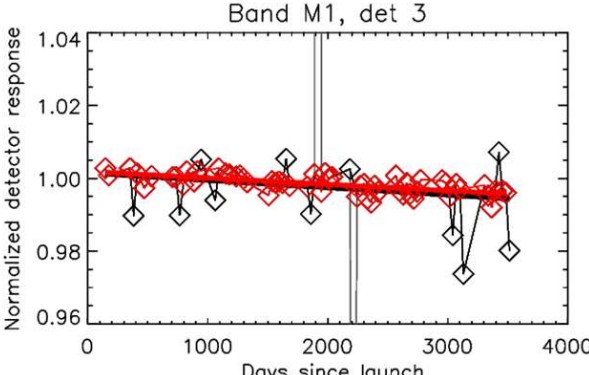

**Figure 17.** Individual lunar detector response fit in band M1 and detector 3. Initially, extreme outliers were filtered by rejecting points larger than ±5 percent, and one standard deviation rejection filtering was also applied to remove the black diamond points. The final linear fit was applied on the remaining data set, indicated as red diamond symbols and a red fitted line.

The filtered lunar detector response differences were calculated for bands M1 to M8 and are shown in Figure 18. The long-term detector response differences were linear in all the RSB detectors and they were fitted using a linear function fit. The detector response differences were observed mostly in bands M1 to M7 that have center wavelengths below one micrometer. The detector response difference in bands M8 to M11 was within the ±0.5% range. In all the VIIRS RSB bands, detector response differences became larger over time, especially in the short wavelength bands below M7. There were a couple of edge detectors (detector 15 and 16) in bands M1 and M2 that showed opposite long-term changes. These responses are compared with the *SD* detector response differences in the following section.

3.4.2. Solar Diffuser Detector Response Differences

The SD detector response differences were derived from the daily averaged SD F-factors from Equation (7). Figure 19 shows SD detector response differences in the VIS-NIR band of RSB. The initial *SD* detector differences were quite large, especially in band M1 compared to the lunar detector differences in Figure 18. Especially for edge detector 16, the starting *SD* detector difference was larger than four percent, whereas the lunar detector difference was less than a 0.5 percent level. This indicated that the SD had some degree of surface non-uniformity from the start of the mission, especially in band M1.

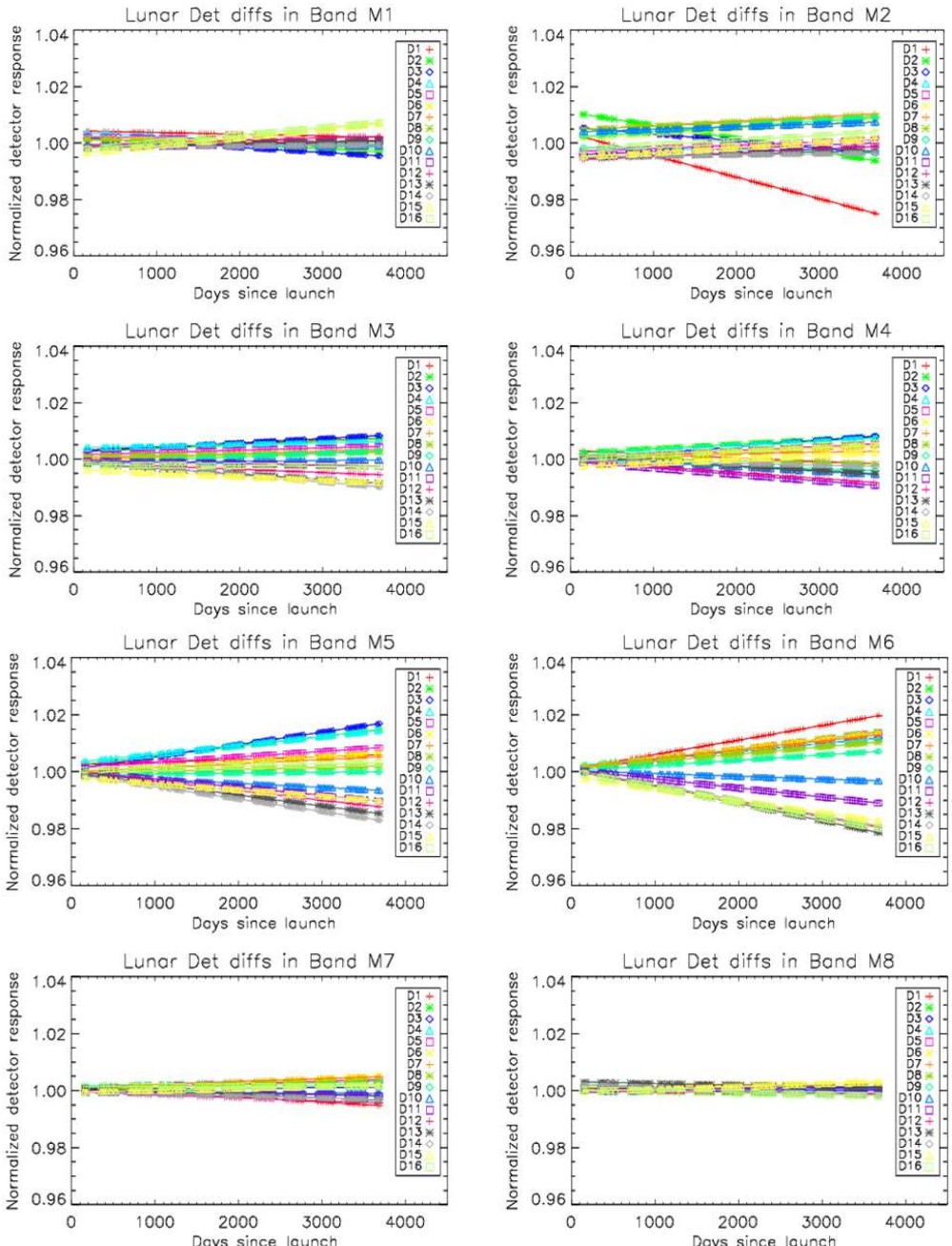

**Figure 18.** Filtered long-term trends of S-NPP VIIRS lunar detector response differences in bands M1 to M8. The lunar detector response differences in bands M8 to M11 were insignificant in that they were mostly within a $\pm 0.5$ percent level. All the lunar collections were forced to be in the high gain state for the dual gain bands.

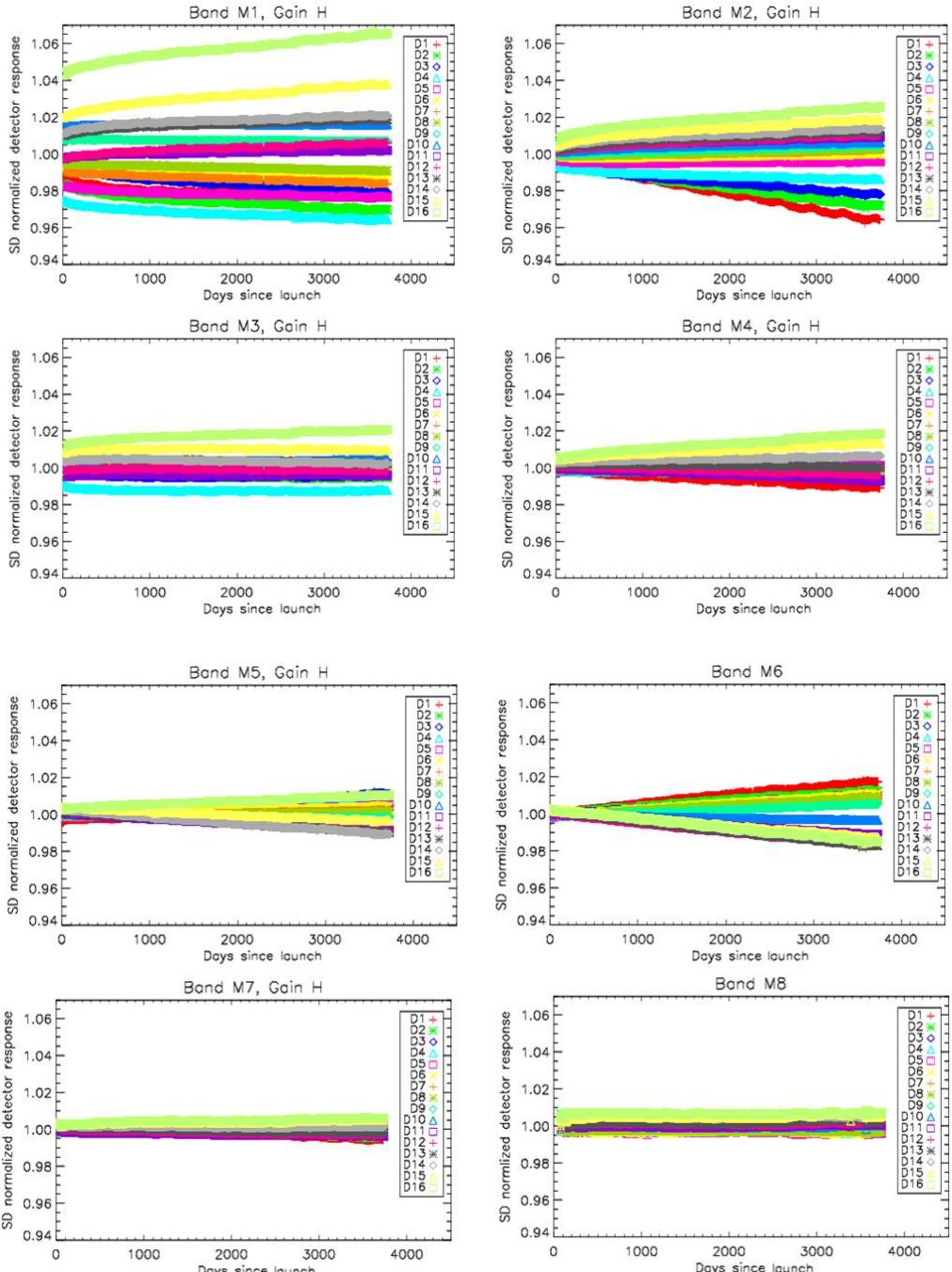

**Figure 19.** Long-term trends of S-NPP VIIRS *SD* detector response differences in bands M1 to M8. The *SD* detector response differences showed larger long-term trends than lunar detector differences, especially in M1 to M4 with the edge detectors. Other bands did not show insignificant differences.

Compared to the lunar detector differences in Figure 18, the *SD* detector differences showed larger long-term slopes caused by the *SD* non-uniformity but not caused by the detector responsivity changes. If there was increasing non-uniformity on the *SD* surfaces at the short side of VIS/NIR bands, especially in the VIIRS detector array direction, it would increase the striping in the SDR products, as shown in Figure 7. To mitigate artifacts from the non-uniform *SD* surface over time, the long-term detector response differences between *SD* and lunar observations were estimated in each detector.

### 3.4.3. Lunar and Solar Diffuser Detector Response Differences

The ratio between normalized *SD* and lunar detector differences was calculated by Equation (8) and is shown in Figure 20. As expected, larger differences were found in recent years at the edge detectors because of the *SD* non-uniformity. Initially, the differences were exponentially increased in the early life, but they were stabilized after 1000 days in bands M1, M2, and M3. These bands showed large differences up to a one percent level in recent years. When considering the differences from detector 1 to 16 in a scan for a uniform surface on the ground, it went up to a two percent level, which caused sudden radiometric level changes on the scan boundaries, as shown in Figure 7. On the other hand, these differences were reduced in the longer wavelength bands from M5 to M11.

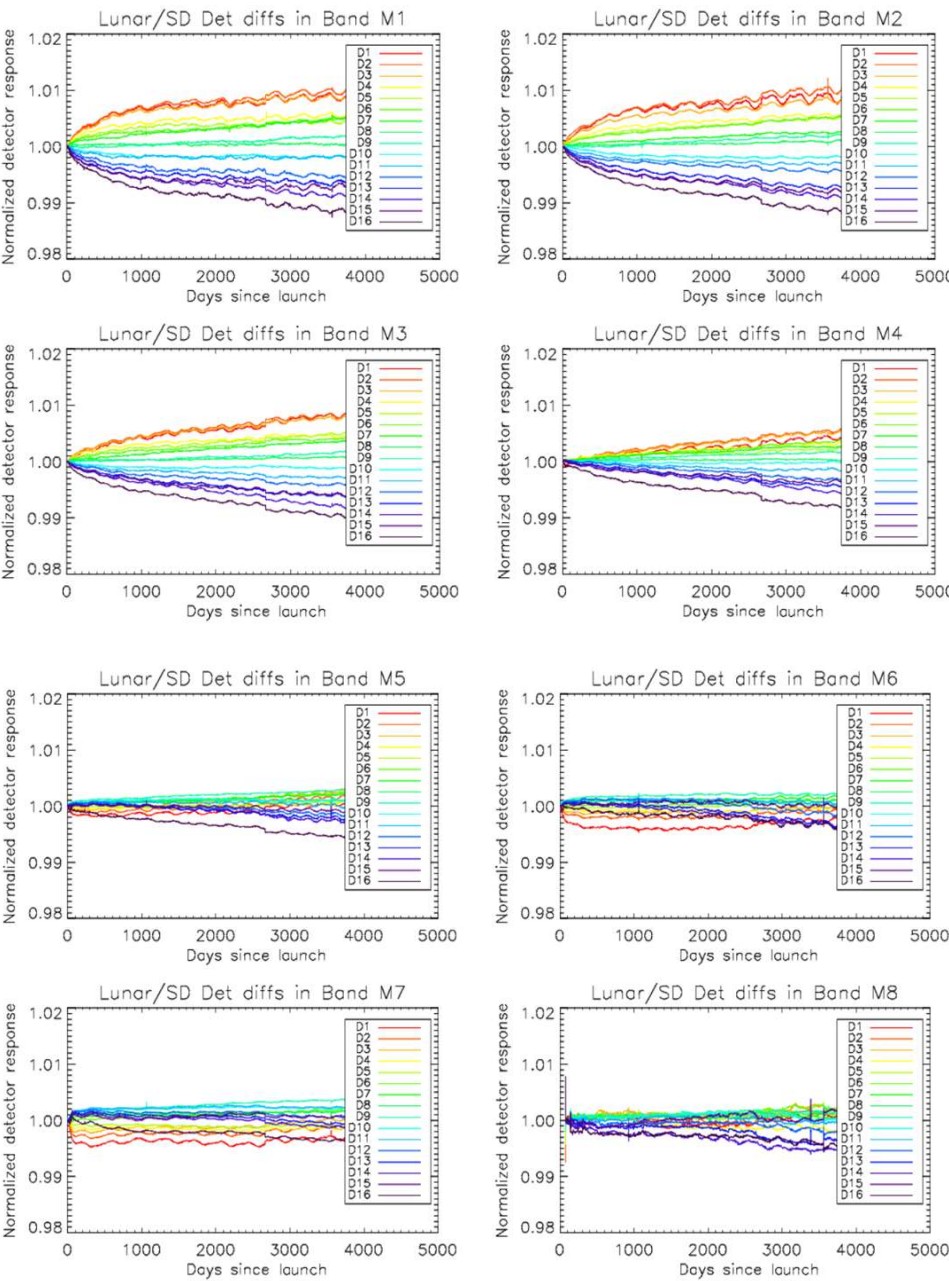

**Figure 20.** S-NPP VIIRS normalized detector response ratio between lunar and SD bands M1 to M8.

### 3.5. Applying Lunar and SD Detector Differences to Current Operational VIIRS Products

Evaluating the remaining scan-based striping is an effective way to test the quality of the on-orbit calibration. The operational products observed by S-NPP VIIRS in short

wavelength bands indicated that there was striping in the scan direction that was gradually increasing [26,27]. By applying the correction factor using Equations (8) and (9), Figure 21 shows before and after the detector striping correction by removing the lunar and *SD* detector response differences caused by the *SD* non-uniformity in the detector array direction. As shown in Figure 21, the striping significantly reduced with the S-NPP M1 image on 5 May 2021 in the Red Sea box shown in Figure 7. For better visualization, a false-color table was used in Figure 13. In the right column of Figure 13, the detector differences in the uniform area were calculated by normalizing the average of the detector response. In the Red Sea case study, there were more than two percent differences in the detector array direction, which was corrected with a 0.5% level after the striping correction. The striping correction was successfully applied on top of the current operational products, and it will be a major improvement of the next version of NOAA's reprocessing.

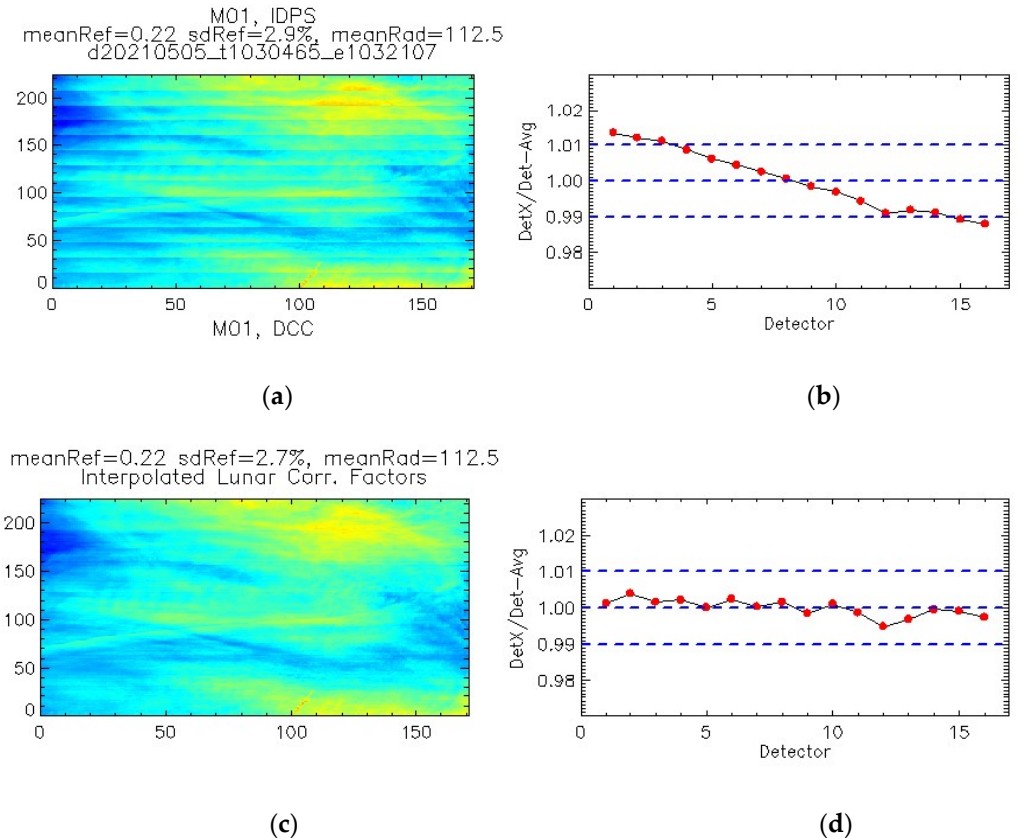

**Figure 21.** An example of S-NPP VIIRS band M1 image of the detector response difference correction over the Red Sea region on 5 May 2021 at 10:30 UTC. The top row subfigures (**a,b**) show the operational S-NPP VIIRS SDR product image and detector difference responses, respectively. The bottom row subfigures (**c,d**) represent corrected image and detector differences.

## 4. Discussion

Over ten years of on-orbit S-NPP VIIRS operations, the lunar and *SD*-based RSB calibration algorithm and methodology has matured and been accepted among several working groups such as the NOAA VIIRS SDR team, the NOAA Ocean Color team, the NASA VIIRS Calibration Support Team (VCST), and the NASA Ocean Biology Processing Group (OBPG) [8,9,20,21,42]. Nevertheless, the sub-percent level accurate on-orbit calibration of S-NPP VIIRS is still challenging because of the long-term difference between the *SD* and lunar F-factors, *SD* F-factor annual oscillations in conjunction with H-factors, and non-uniformity of the *SD* surface that caused a two-percent difference in the short wavelength bands. To validate on-orbit *SD* F-factors, lunar F-factors provided a long-term trend in each RSB as an alternative source of calibration. Monthly lunar F-factors successfully validated

the long-term trends of the primary *SD* F-factors; however, the standard deviations of difference between the moon and *SD* F-factors were mostly around one-percent levels, as shown in Table 2. On top of these differences, there were growing long-term trend differences between the *SD* and lunar F-factors, especially in bands M1 to M4. The larger differences were observed in the early life of S-NPP VIIRS, which introduced different analyses and correction methodologies on the initial states from 2012 to 2014. NASA VCST applied these initial difference corrections into their Collection 2 products, and the NOAA VIIRS team also developed a comprehensive correction based on lunar, DCC, and SNO trends for reprocessing version 2 [23,43]. One of the major inputs of NASA's correction factor was the lunar F-factor.

The annual oscillations in the *SD* F-factors were caused by the imperfect SDSM sun transmittance ($\tau_{SDSM}$ in Equation (3)), and a new version $\tau_{SDSM}$ was applied to the S-NPP version-2 reprocessing [14,23]. With the new $\tau_{SDSM}$, the one-percent levels of annual oscillation patterns were significantly reduced in the short wavelength RSB bands from M1 to M4. Over the lifetime of S-NPP, the *SD* DN values gradually decreased overtime, which are in the denominator part of F-factor equation. As a result of this, the *SD* F-factor oscillations have increased over time, especially in the short wavelength bands (M1~M4) in Figure 16.

On the other hand, the lunar F-factor oscillations were caused by the residual lunar libration effects from the GIRO lunar irradiance model [42]. Further investigation is needed to reduce the annual oscillation of the lunar F-factors.

From near-nadir observations over a homogeneous site such as Libya 4 desert and DCC, the VIIRS SDR product showed increasing calibration differences among the detectors [26,27,43]. Instead of using the SDR products, these detector differences were systematically estimated using the detector response differences between the *SD* and lunar observations. These differences could be caused by the viewing angle effects from *SD* to VIIRS detectors. In addition, it was assumed that the non-linear effects could be the source of response difference, because of the signal level differences between moon and SD observations. However, non-linear effects were not the source of the differences because the DCC estimations, which were at the top of the dynamic range, also provided similar results to the moon-based trends [44]. A set of correction factors was successfully derived using *SD* and lunar detector response differences to mitigate the string caused by the *SD* non-uniformity. The correction factors from the lunar and *SD* observation were almost identical to the DCC-based estimations, which proved the validity of the current methodology.

## 5. Conclusions

Over the last ten years, S-NPP VIIRS has performed well and successfully provided EV observations. The *SD* has provided a primary source of on-orbit calibration as a transferring radiometer with the measured BRDF function from the extensive prelaunch calibration. Due to increased surface roughness from the UV portion of solar illumination, the reflectance of *SD* has been degraded, and it was measured and compensated to the on-orbit calibration called *SD* F-factors. However, there were small long-term trend differences compared to the lunar F-factors in bands M1 to M4. The *SD* F-factors have been successfully validated by using monthly lunar calibration coefficients (called lunar F-factors) within two percent standard deviation compared to SD trends over a decade.

Alternatively, the long-term normalized *SD* and lunar detector response ratios were derived and applied to mitigate the scan striping, especially in the short wavelength bands. A systematic approach using *SD* and lunar collections successfully evaluated the non-uniformity of *SD* degradation over time in the detector array direction. The derived long-term correction factors were validated by independent DCC-based results. Obviously, lunar observations played a crucial role for S-NPP VIIRS on-orbit calibration in conjunction to the *SD* calibration. The long-term lunar trends provided an independent source of calibration to validate or correct the relative calibration differences from the *SD* calibration. Especially, the lunar detector dependent observations were used to find out the

non-uniform *SD* degradation over time, which was the source of a scan-based string in the images of short wavelength bands. It should be noted that the on-orbit lunar calibration is not an option but a necessary investment for future remote sensing instruments to ensure the long-term radiometric accuracy of its related scientific products.

**Author Contributions:** T.C. collected and prepared the lunar data, developed the processing code, analyzed the results and wrote the manuscript. C.C. provided technical guidance and supervision. X.S. provided conceptualization and formal analysis. W.W. provided validation results. All authors have read and agreed to the published version of the manuscript.

**Funding:** This research was funded by the PROTECH contract awarded to Global Science & Technology (GST) by NOAA/National Environmental Satellite, Data, and Information Service (NESDIS).

**Conflicts of Interest:** The authors declare no conflict of interest.

**Disclaimer:** The scientific results and conclusions, as well as any views or opinions expressed herein, are those of the author(s) and do not necessarily reflect those of NOAA or the Department of Commerce.

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
