# Peer review of "S-NPP VIIRS Lunar Calibrations over 10 Years in Reflective Solar Bands (RSB)"

_remotesensing, doi:10.3390/rs14143367_

Round 1

Reviewer 1 Report

I think the uniqueness of this paper is the evaluation of independent detector degradation factor from Solar Diffuser (SD) and Lunar calibration data.

In general, diffuser is more uniform than lunar image because lunar reflectance difference at the different point is twice.

However, the author extracted the small diffuser non-uniformity using full disk image of moon.

To understand this methodology more clearly, the following points should be described and discussed.

Q1) From Figure 9, there is an assumption the overlap (or underlap) between scans (or within scan) must be identical between detectors. The effect of different sampling between detectors may be similar.

Q2) The incident angle difference to the diffuser of detector view vector has similar effect.

Q3) The signal level from lunar and diffuser might be different. The non-linearity effects because of different signal level is similar.

The following points may be typographical error.

Q4) Figure 2 in LINE-292 is Figure 8 ?

Q5) Figure 5 in LINE-299 is Figure 8 ?

Q6) Figure 8 in LINE-300 is Figure 9 ?

Q7) Poor explanation of Figure 16. Marks are lunar results? Lines are SD results?

Author Response

Thank you for the excellent suggestions. We tried out best to respond your comments. The line numbers are based on the revised version.

Comments and Suggestions for Authors

I think the uniqueness of this paper is the evaluation of independent detector degradation factor from Solar Diffuser (SD) and Lunar calibration data.

In general, diffuser is more uniform than lunar image because lunar reflectance difference at the different point is twice.

However, the author extracted the small diffuser non-uniformity using full disk image of moon.

To understand this methodology more clearly, the following points should be described and discussed.

Thank you for recognizing the value of our paper.

Q1) From Figure 9, there is an assumption the overlap (or underlap) between scans (or within scan) must be identical between detectors. The effect of different sampling between detectors may be similar.

Accepted. This is an excellent comment that we missed our basic assumption. We added explanations in the manuscript below Figure 9. (near line 312)

Q2) The incident angle difference to the diffuser of detector view vector has similar effect.

Accepted. We added a sentence in the discussion section (near line # 619).

Q3) The signal level from lunar and diffuser might be different. The non-linearity effects because of different signal level is similar.

This is another excellent comment. Initially, we also doubted that the source of the detector response differences could be caused by the non-linearity issues since the moon responses were in the low end whereas SD responses were located at the middle range. But the DCC based results were very similar (or almost identical) results and DCC results were from the high level of dynamic range. These DCC results indicated that the detector response differences were not caused by the detector non-linearity. We added explanations near line #620 in the discussion section. A reference [44] is also added for the DCC results.

The following points may be typographical error.

Q4) Figure 2 in LINE-292 is Figure 8 ?

Corrected.

Q5) Figure 5 in LINE-299 is Figure 8 ?

Corrected.

Q6) Figure 8 in LINE-300 is Figure 9 ?

Corrected.

Q7) Poor explanation of Figure 16. Marks are lunar results? Lines are SD results?

Accepted. Added a sentence in the figure caption.

Reviewer 2 Report

The paper reviews the NOAA team’s work on the S-NPP VIIRS RSB on-orbit calibration, showing the H- and SD F-factors equations and results. Additionally, the authors use the lunar F-factors to correct the SD F-factor long term trend and detector-to-detector difference issues. I have two general concerns on the paper.

One concern is that I am unsure how the authors applied the differences to correct the SD F-factors in real-time, namely, a formula is needed to extend in time what has been observed for the correction. If a formula is used to apply the correction in real-time, I’d like to see the formula in the paper.

Another concern is that the H-factors for the SWIR bands are not clearly mentioned in the paper. Does this mean that the H-factors for the SWIR bands are assumed to be one? How does the NASA team handle the SWIR band H-factors?

Other comments are listed below.

91-92 should be: RTA and SDSM SD viewing angles

141 Eq. (2) misses the RSB detector relative spectral response (RSR) function and Esun should be part of the RSR wavelength integral.

143  Angle theta(inc) should be the solar angle with respect to the SD surface not the SD screen.

158-159 DC is the unit for an SDSM detector count and DN is the unit for an RSB detector count. The two units are not the same.

257 Eq. (5) needs a reference that actually derives the equation.

260 GIRO should be spelled out when used for the first time in the paper.

289-291 “degradation in different detectors” is unclear: the RSB detector gain or the SD degradation.

Fig. 12 The meaning the of the y-axis should be clearly labeled in the figure.

421-426 The H-factor has a gap before and after the Feb. 24, 2019 event and the dn(SD) for an RSB detector has the same gap, meaning that the F-factor is more or less the same across the event. Why is there a spike in the F-factor in Fig. 14? (The authors acknowledge in the paper that the event caused SD screen blocking.) Is the spike in the F-factor from the auto-cal?

443 “standard level” -> “standard deviation level”?

589-590 The NASA work needs a reference citation.

594-598 The magnitude of the F-factor annual oscillation seems to get larger with time, shown by Fig. 16.  So how a fix in the SDSM screen can resolve the oscillation. Are there other reasons for the annual oscillations?

Author Response

The paper reviews the NOAA team’s work on the S-NPP VIIRS RSB on-orbit calibration, showing the H- and SD F-factors equations and results. Additionally, the authors use the lunar F-factors to correct the SD F-factor long term trend and detector-to-detector difference issues. I have two general concerns on the paper.

One concern is that I am unsure how the authors applied the differences to correct the SD F-factors in real-time, namely, a formula is needed to extend in time what has been observed for the correction. If a formula is used to apply the correction in real-time, I’d like to see the formula in the paper.

Currently, we are not applying this correction in real-time. The correction will be considered for the future version of S-NPP VIIRS reprocessing.

Another concern is that the H-factors for the SWIR bands are not clearly mentioned in the paper. Does this mean that the H-factors for the SWIR bands are assumed to be one? How does the NASA team handle the SWIR band H-factors?

Correct. The SWIR band H-factors are set to be one for M8 to M11. NASA team extrapolated M8 to M11 wavelength from an empirical model using SDSM detectors from 5 to 8 at each SDSM collection [25], which gave small portion of degradation over time. For NOAA products, the SWIR band degradation was applied to reprocessed data [23]. We added a sentence near line # 363. “For SWIR band calibration, the H-factors are assumed to be one for current operational products.”

Other comments are listed below.

91-92 should be: RTA and SDSM SD viewing angles

Accepted and corrected.

141 Eq. (2) misses the RSB detector relative spectral response (RSR) function and Esun should be part of the RSR wavelength integral.

Accepted and added RSR and integral over the wavelengths.

143  Angle theta(inc) should be the solar angle with respect to the SD surface not the SD screen.

Accepted and corrected.

158-159 DC is the unit for an SDSM detector count and DN is the unit for an RSB detector count. The two units are not the same.

Accepted and added a sentence about the differences between DC and DN counts near line 162.

257 Eq. (5) needs a reference that actually derives the equation.

Accepted and added a reference for the equation.

260 GIRO should be spelled out when used for the first time in the paper.

Accepted and spelled out.

289-291 “degradation in different detectors” is unclear: the RSB detector dgain or the SD degradation.

Accepted. We revised the sentence to “there are higher chances of having differences of SD degradation in different detector positions.”

Fig. 12 The meaning the of the y-axis should be clearly labeled in the figure.

Accepted and corrected. The referring location of Fig.12 was wrong.  The LTAN profile is not shown in Figure 12. But the range of solar azimuth angle to the SD was altered. The y-axis caption was changed to ‘SD Azimuth Angle [Degrees]’ and an arrow is added in the figure.

421-426 The H-factor has a gap before and after the Feb. 24, 2019 event and the dn(SD) for an RSB detector has the same gap, meaning that the F-factor is more or less the same across the event. Why is there a spike in the F-factor in Fig. 14? (The authors acknowledge in the paper that the event caused SD screen blocking.) Is the spike in the F-factor from the auto-cal?

Accepted. This is an excellent comment. The sudden change of SD DN (F-factor) and SDSM DC (H-factor) changes caused sudden spikes. When the H-factors were derived, there were slight time-delays (linear change over time) from an averaging filter in the H-factor around Feb. of 2019 in Figure 11. In this situation, the F-factors were calculated non-filtered DN from the SD that included immediate signal changes after the anomaly event. As a result, there were some time gap of the H-factor to catch-up the change. These asynchronous H-factor caused the spikes in the F-factors. These spikes were observed in the auto-cal. To explain these sudden spikes, we added a sentence, “Because of the time-delays in the H-factor filtering (in Figure 11) around February 24th 2019, there were sudden F-factor spikes with the immediate SD DN signal changes until the H-factor went back to the nominal trends.”

443 “standard level” -> “standard deviation level”?

Accepted and corrected as suggested.

589-590 The NASA work needs a reference citation.

Accepted. The references of NASA’s work were placed at the next sentence. We replaced these references to the right place.

594-598 The magnitude of the F-factor annual oscillation seems to get larger with time, shown by Fig. 16.  So how a fix in the SDSM screen can resolve the oscillation. Are there other reasons for the annual oscillations?

This is a good observation. The S-NPP VIIRS F-factor oscillations are caused by the inadequate application of the SDSM Sun transmittance function (). Our previous study [14] showed that the old version of the  produced consistent one percent annual oscillations in the H-factors. The F-factor is linearly related to H-factor and calculated inverse of SD DN as explained in Equation (2). Over the lifetime of S-NPP, the SD DN values are gradually decreased overtime which is in the denominator part of F-factor. As a result of it, the oscillations levels are getting increase over time especially in the short wavelength bands (M1-M4).

By applying the new in the H-factor equation, annual oscillations in the H-factors were reduced by one percent in the reprocessed product (version 2) [23]. The initial had inadequate transition especially in the solar azimuth dimension.

The annual oscillations can be cause by the inadequate prelaunch measurements that are applied in the on-orbit calibration equations. For example, the initial version of the function caused 1 percent of oscillations. Actually, we can not remove all the annual oscillations in the on-orbit calibration as long as there are errors in the calibration.

We added our responses of your comment as shown below.

“Over the lifetime of S-NPP, the SD DN values have gradually decreased overtime which are in the denominator part of F-factor equation. As a result of it, the SD F-factor oscillations have increased over time especially in the short wavelength bands (M1-M4) in Figure 16.”